# Spatiotemporal distribution of power outages with climate events and social vulnerability in the USA

Vivian Do [1], Heather McBrien[1], Nina M. Flores[1], Alexander J. Northrop [2], Jeffrey Schlegelmilch[3], Mathew V. Kiang [4] & Joan A. Casey [1,5] ✉

Power outages threaten public health. While outages will likely increase with climate change, an aging electrical grid, and increased energy demand, little is known about their frequency and distribution within states. Here, we characterize 2018–2020 outages, finding an average of 520 million customer-hours total without power annually across 2447 US counties (73.7% of the US population). 17,484 8+ hour outages (a medically-relevant duration with potential health consequences) and 231,174 1+ hour outages took place, with greatest prevalence in Northeastern, Southern, and Appalachian counties. Arkansas, Louisiana, and Michigan counties experience a dual burden of frequent 8+ hour outages and high social vulnerability and prevalence of electricity-dependent durable medical equipment use. 62.1% of 8+ hour outages co-occur with extreme weather/climate events, particularly heavy precipitation, anomalous heat, and tropical cyclones. Results could support future large-scale epidemiology studies, inform equitable disaster preparedness and response, and prioritize geographic areas for resource allocation and interventions.

As climate change intensifies, the power grid ages, and energy demand from population growth increases, power outages will likely increase[1]. In 2020, US electricity customers faced slightly over 8 h of electricity interruptions on average – the highest on record—primarily driven by major events such as hurricanes, wildfires, and snowstorms[2]. When outages occur, human health suffers[3]. The United States Federal Emergency Management Agency (FEMA) identifies the power grid among Community Lifelines, which are fundamental services that society needs in order to operate[3]. Documented health effects include carbon monoxide poisoning from improper generator use, anxiety, stress, and exacerbation of existing cardiovascular and respiratory conditions[4]. Because outages can prevent the use of temperature-controlling devices, risk of hypothermia and heatstroke can increase when outages occur during extreme cold spells and heatwaves[5].

Moreover, outages can lead to acute food insecurity when refrigerators lack power[6], fear related to personal safety[7], and economic losses in commercial and industrial sectors[8].

Power outages represent acute health hazards for certain vulnerable groups. Those using electricity-dependent durable medical equipment (DME), such as oxygen concentrators, infusion pumps, and mobility devices rely on electricity to maintain their health[9,10]. Others vulnerable to power outages include under-resourced communities and historically marginalized groups. Pathways include disrupted hourly employment, older and less-insulated housing stock resulting in dangerous indoor temperatures, lack of access to cooling facilities, and a higher burden of underlying chronic diseases sensitive to extreme temperatures[11,12]. Other historically marginalized groups may face more adverse health outcomes following outages, or worse

[1]Department of Environmental Health Sciences, Columbia University Mailman School of Public Health, New York, NY, USA. [2]Vagelos College of Physicians and Surgeons, Columbia University, New York, NY, USA. [3]National Center for Disaster Preparedness at the Columbia Climate School, Columbia University, New York, NY, USA. [4]Department of Epidemiology and Population Health, Stanford University School of Medicine, Stanford, CA, USA. [5]Department of Environmental and Occupational Health Sciences, University of Washington School of Public Health, Seattle, WA, USA. ✉e-mail: jacasey@uw.edu

outage exposures[4,13]. Communities with a higher proportion of Hispanic/Latino residents may experience longer outages following hurricane and winter storm-related events[14–16].

Although technical problems such as equipment failure and supply shortages can cause outages, severe weather events, which can physically damage the grid, are major drivers[4,17]. Power distribution infrastructure such as transmission lines are also vulnerable to extreme environmental events such as high temperatures, wildfires, and floods[18]. From 2000–2021, storms and severe weather caused 83% of large-scale outages affecting at least 50,000 customers in the US[1]. There is limited work on the link between environmental events and smaller-scale (but more frequent) outages.

No standard method exists to measure power outages of health relevance, making it challenging to compare outage events[19]. Most population health power outage studies have focused on a single, large event, such as Hurricane Sandy in 2012 or Hurricane Irma in 2017[4]. These studies generally did not measure customers without power and instead used the timing and location of the disaster as a proxy for outage exposure[4]. Presently, no national power outage datasets exist at the temporal or spatial resolutions necessary for health studies. Here, we address this gap by creating relative and absolute measures to characterize power outages across the US from 2018–2020 by hour at the county-level. The relative metric accounts for population size, while the absolute metric identifies counties with the largest count of customers without power. Both metrics provide important information about which counties to prioritize for intervention and resource allocation, especially in the context of social and medical vulnerabilities. In secondary analyses, we determine the overlap between weather events occurring on the same day as 8+ hour (a medically-relevant duration with potential health consequences) county-level outages and clustering of counties experiencing high outage burden and high social vulnerability.

Between 2018–2020, we identified 231,174 1+ hour outages and nearly 17,500 8+ hour (medically relevant) outages at the county-level. 62.1% of the 8+ hour outages co-occur with an extreme weather or climate event and 8+ hour outages are 3.4x more common on days with a single event and 10x more common on days with multiple events. Outages are more common in the Northeast, South, and

Appalachia. Clusters of counties in Arkansas, Louisiana and Michigan experience a dual burden of high outage exposure and high social and medical vulnerability.

## Results

The study included 2447 counties (77.9% of US counties) of which 2038 (83.3%) had 2+ years of reliable data from 2018–2020 after data quality and reliability checks (Fig. 1). These counties experienced a median of 60 (IQR = 97) 1+ hour outages and 2 (IQR = 5) 8+ hour outages each year. Between 2018–2020, over 70% of included counties experienced at least one 8+ hour outage and a total of 231,174 1+ hour outages and 17,484 8+ hour outages occurred (Table 1). Medically relevant 8+ outages happened more often during the summer than the winter and peaked during late spring and mid-summer (Fig. 2). 8+ hour outages typically had an onset around 6 PM with a range of 3 PM-8 PM (Fig. 2), coinciding with peak electricity use, and this pattern was especially prominent in the South (Supplementary Fig. 2). In the ten states with the most 8+ hour outages, outages were more common in April and October (Supplementary Fig. 3). Our analysis did not cover all counties; states with the highest percent of counties missing all years of data were Montana (98.2%), Alaska (93.1%), and Utah (72.4%) (Supplementary Table 1).

### Characterizing outage events and customers without power

In our 2447 study counties, the highest average counts of 8+ hour outages occurred in the South, Maine, Michigan, and Appalachia (Fig. 3a). Figure 3 also illustrates the pattern of data availability and reliability, with a higher prevalence of counties with 3 years of complete and reliable data on the East Coast (darker shading) and a low prevalence of availability and reliability in the middle of the U.S (lighter or no shading). The states with the highest annual average counts of 8+ hour outage events were Louisiana (n = 553), Texas (n = 527), Michigan (n = 447), Mississippi (n = 381), and North Carolina (n = 372). When we created deciles of county-level annual average 8+ hour outage counts, the states with the highest number of counties in the top outage decile were Michigan (n = 32) and Louisiana (n = 29) (Supplementary Table 2).

The spatial distribution of 1+ hour outages generally mirrored that of 8+ hour outages but extended to additional counties; outages were

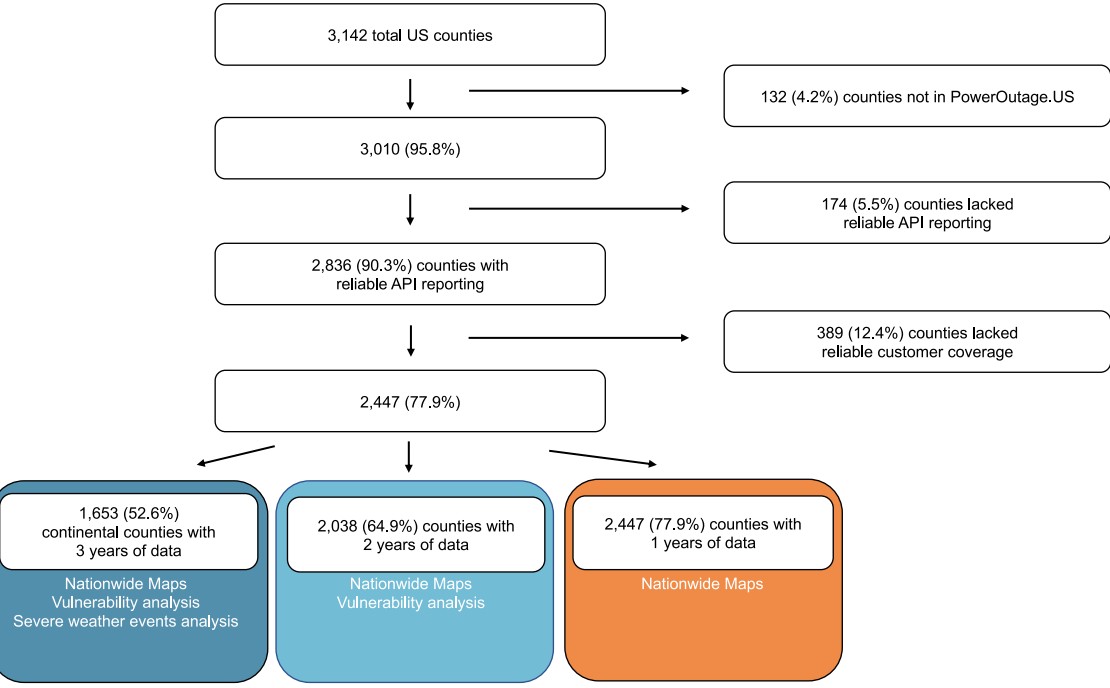

**Fig. 1 | Flowchart of US counties included in the study of power outages, 2018–2020.** We state the subsets of counties used for specific analyses in colored boxes. Power outage data was purchased from PowerOutage.us.

**Table 1 | Summary statistics of 8+ hour outages and 1+ hour outages among counties with 2+ years of reliable data**

|  | Annual average 8+ hour outage | Annual average 1+ hour outage |
|---|---|---|
| Counties included in analysis, N | 2038 | 2038 |
| Counties with ≥1 outage, N (%) | 1436 (70.5) | 1530 (75.1) |
| Total outage count | 17,484 | 231,174 |
| Median (IQR) county-level outage count | 2 (5) | 60 (97) |
| Min county-level outage count | 0 | 0 |
| Max county-level outage count | 35 | 414 |

Summary statistics refer to the average yearly totals per county, which is the total of outage types per county that are averaged across the study period (2018–2020). Power outage data was purchased from PowerOutage.us.

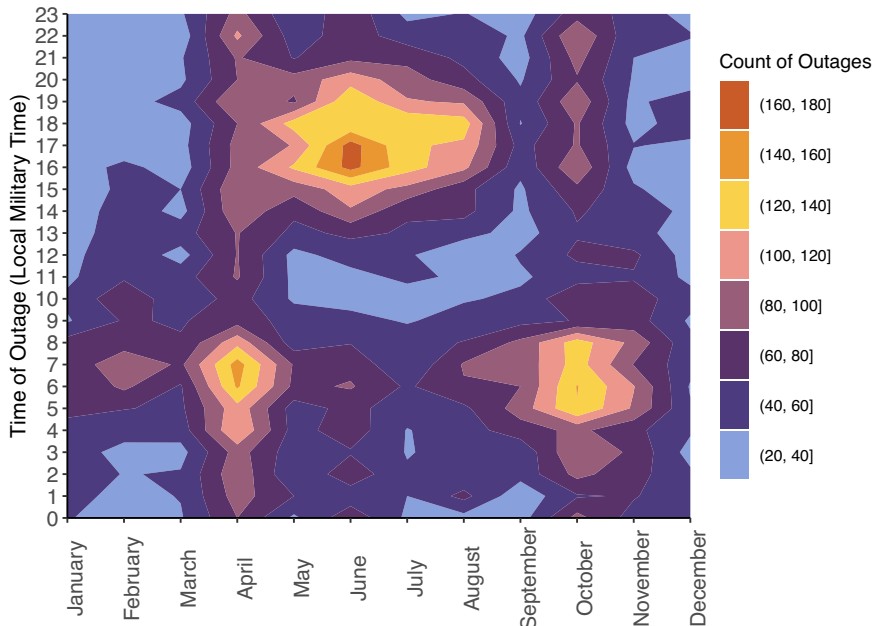

**Fig. 2 | Contour plot of the counts of 8+ hour outages nationwide according to start time of outage per month.** The x-axis indicates start month, the y-axis indicates the start hour of an 8+ hour outage in local military time, and the colors indicate the total count of outages for that month and start hour. Power outage data was purchased from PowerOutage.us.

concentrated in the South, the Northeast, Appalachia, and parts of California. The lowest counts of 1+ hour outages appeared in the Midwest (Fig. 3b). The top 5 states for highest annual average counts of 1+ hour outages were Texas (N = 11,504), Georgia (N = 10,609), Louisiana (N = 7826), Mississippi (N = 7188), and Alabama (N = 6240). Like 8+ hour outages, Texas, Louisiana, and Mississippi had the largest number of counties in the top outage decile of 1+ hour outage counts with 37, 28, and 22 counties involved, respectively (Supplementary Table 2).

The absolute annual average total customer hours without power partially reflected population size, with a high number of customers without power along the Gulf Coast, the Northeast Coast, and parts of the Pacific Northwest (Fig. 4a). Outages resulted in an annual average of 5.2 million customer-hours without power across 2447 study counties. Some counties in Southern states such as Louisiana, as well as throughout Appalachia, and the Northeast consistently experienced both high counts of outage events and high total customer-hours out. Overall, the state of Louisiana led with an annual average of >52 million customer-hours out, followed by North Carolina (38.3 million customer-hours out), California (30.3 million customer-hours out), Texas (30 million customer-hours out), and New York (28.8 million customer-hours out). Counties with the highest annual average total hours of customers without power were Calcasieu, LA (10.0 million), Los Angeles, CA (7.8 million), Fairfield, CT (7.6 million), Davidson, TN (7.4 million), and Jefferson, LA (6.2 million) (Supplementary Table 3). When accounting for county-level customers, counties where the

average customer experienced a high annual average of hours without power (107+ hours) concentrated around the Gulf Coast and the Northeast, particularly Maine (Fig. 4b). Orleans (NY) led, where the average customer experienced 251.4 h, or over 10 full days, without power each year.

**Severe weather and climate events and 8+ hour outages**

We explored the role of severe weather and climate events in county-level 8+ hour power outages by determining days where isolated and multiple events co-occurred with outages in the continental counties with 3 years of reliable data (n = 1653). Approximately 13% of county-days (n = 22,793) had an 8+ hour outage, 62.1% (n = 14,156) of these county-days co-occurred with one or more weather or climate events (Table 2). Because multiple weather and climate events can occur in the same county on the same day, we divided analysis into county-days with isolated events and county-days with multiple events. 8+ hour outages were 3.4x more common on county-days with an isolated event and 10x more common on days with multiple events, compared to county-days without any severe weather or climate event. Every severe weather or climate event we evaluated, except anomalous cold alone, were related to increased occurrence of 8+ hour outages. Tropical cyclone county-days, while not particularly common (0.2% of outage county-days), were much more likely (13.7x) to co-occur with an 8+ hour outage than county-days without any event. When tropical cyclones happened with other severe weather or climate events the

**a**

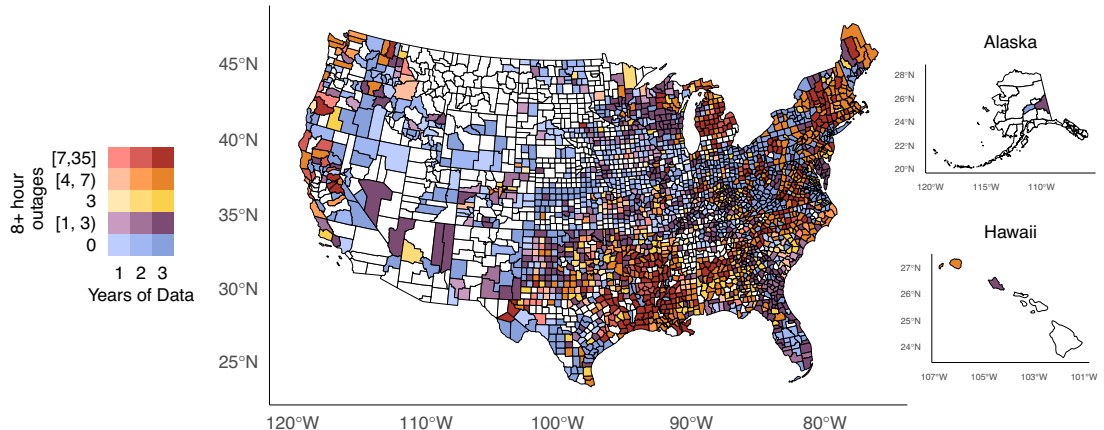

**b**

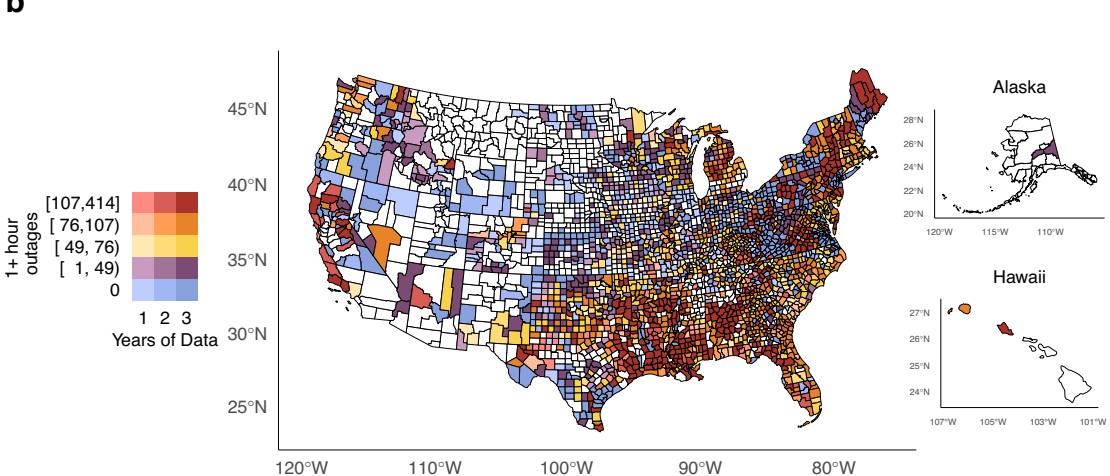

**Fig. 3 | County-level yearly average outage events lasting 8+ h and 1+ hour in 2447 counties with 1 + years of reliable data.** Counties shaded in white lacked any reliable data. **a** Geographic distribution for county-level yearly average of 8+ hour outage events. **b** Geographic distribution for county-level yearly average of 1+ hour outage events. Power outage data was purchased from PowerOutage.us and county basemaps were obtained from the usmap R package version 0.6.1.

likelihood of an 8+ hour outage was even greater. For example, county-days with heavy precipitation and a tropical cyclone (representing 3.0% of total outage county-days) were 37.6x more likely to have an 8+ h outage than county-days without an event. County-days with simultaneous heavy precipitation, anomalous heat, and a tropical cyclone were 51.6x more likely to have an 8+ h outage.

While tropical cyclones seemed to confer the greatest increase in 8+ hour outages, other events were more common. On county-days when 8+ hour outage occurred in conjunction with a single weather or climate event, 75.2% ($n = 8507$) happened with heavy precipitation. When heavy precipitation county-days occurred, 8+ hour outages happened 4.7x more frequently than county-days without an event. Over a third ($n = 2846$) of days when a county faced 8+ hour outages co-occurred with multiple weather or climate events. The most common multiple events co-occurring with 8+ hour outages were heavy precipitation and anomalous heat (32.2%), heavy precipitation and tropical cyclones (23.9%) and heavy precipitation and lightning (22.6%). Other multiple event types occurred on the remaining 606 county-days with an 8+ hour outage and a multiple event (Supplementary Table 4).

Seasonal and geographic patterns of county-level 8+ hour outage-days and severe weather and climate type emerged. For isolated events nationwide, heavy precipitation and snowfall predominated in the winter months, anomalous heat in the summer months, and tropical cyclones and wildfires played a role between July and November (Fig. 5a). Most days with 8+ hour outages in the Northeast, Midwest, and South happened simultaneously with heavy precipitation. Snowfall contributed more 8+ hour outages in the winter in the West, and wildfires made up nearly 75% of co-occurring events in the West in September and October (Supplementary Fig. 4a). Regarding outage county-days co-occurring with multiple events, during summer months the heavy precipitation-anomalously hot temperatures combination happened most often. During the fall, heavy-precipitation and tropical cyclone predominated, while snowfall and anomalous cold was the most common during winter and into spring when heavy precipitation and lightning took over as the most frequency combination (Fig. 5b).

### High 8+ hour outage exposure and vulnerability factors

Our vulnerability analyses relied on the 2038 counties with 2+ years of reliable data. We categorized counties with SVI indices in the 4th

a

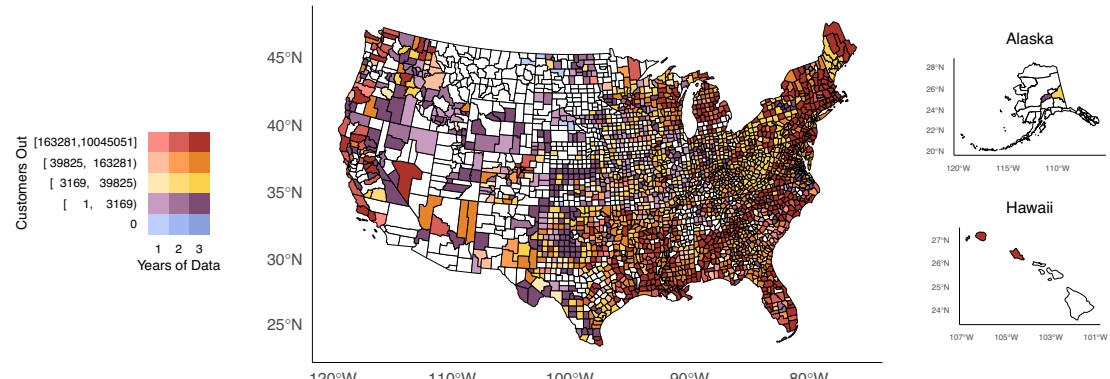

b

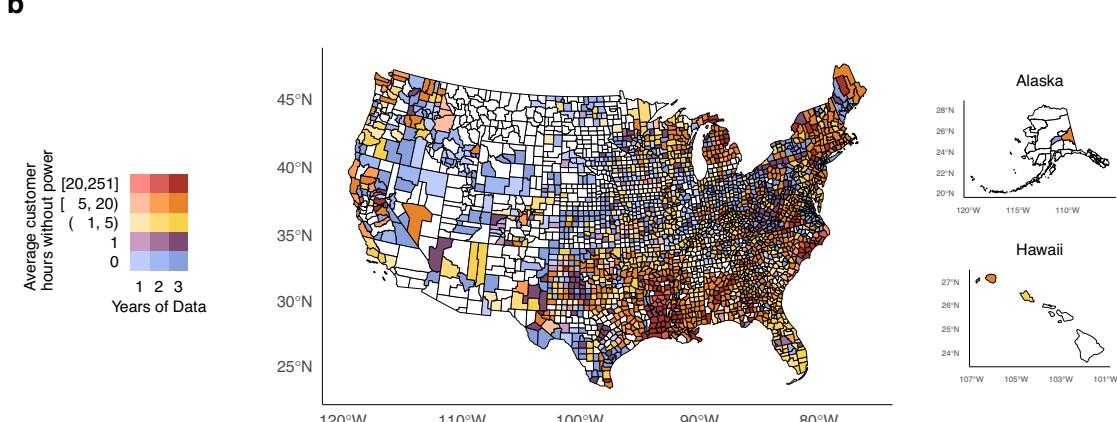

**Fig. 4 | County yearly averages of customers without power.** Counties shaded in white lacked any reliable data. **a** Average total customer hours without power. **b** Average total customer hours without power per customer. Panel **b** can be interpreted as the county-level annual average hours without power that an average customer in that county experienced. Power outage data was purchased from PowerOutage.us and county basemaps were obtained from the usmap R package version 0.6.1.

quartile (range: 0.77–1) as high vulnerability. We observed high SVI in parts of the West Coast, the Southeast, and throughout the South (Supplementary Fig. 5a). Counties in the highest SVI quartile experienced an annual median of 3 (IQR = 5) 8+ hour outages and 80 (IQR = 83) 1+ hour outages compared to an annual median of 1 (IQR = 4) 8+ hour outages and 40 (IQR = 73) 1+ hour outages for counties in the lowest SVI quartile (Table 3, Wilcoxon Rank Sum *p*-value < 0.01). Outages for counties with the top SVI quartile occurred most often in April, June, and October (Fig. 6a). This trend persisted across census regions, driven by the South (Supplementary Fig. 6).

DME analyses also included the 2038 counties with 2+years of reliable data. The 4th quartile for our DME metric ranged from 74–478 Medicare DME users per 1000 Medicare beneficiaries and most counties in this category were in the Mountain West, parts of the South, and Appalachia (Supplementary Fig. 5b). Counties in the highest DME use prevalence quartile experienced a yearly median of 1 (IQR = 3) 8+ hour outages and 42 (IQR = 82) 1+ hour outages, significantly lower counts compared to the other three quartiles (Fig. 6b, Wilcoxon Rank Sum test *p*-value < 0.01).

We used bivariate LISA analysis with a false discovery rate method to identify clusters of counties with a dual burden of high 8+ hour power outage counts and high SVI (4th quartile) or high DME use prevalence (4th quartile). We identified 63 counties in

7 states with both high 8+ hour outages and high SVI. These high-high county clusters were largely concentrated in Louisiana (*n* = 26), Mississippi (*n* = 12), Arkansas (*n* = 9), and Michigan (*n* = 8) (Fig. 7a). Among high outage-high SVI counties compared to all others, the components of SVI contributing to a high SVI score were higher percentages of "racial and ethnic minority" individuals (40.4% vs. 24.0%), individuals living below 150% poverty (34.0% vs. 24.3%), and those living in mobile homes (20.1% vs. 12.4%). *T*-tests also showed that these differences were statistically significant (*p*-value < 0.05) (Supplementary Table 4). For Medicare DME use prevalence, there were 38 counties in 8 states with high annual average counts of 8+ hour outages and high DME use prevalence. States with the most high-high clusters were Arkansas (*n* = 12), Michigan (*n* = 10), and Louisiana (*n* = 9) (Fig. 7b).

## Discussion

This study used hourly county-level data between 2018–2020 from 2447 US counties, covering 73.7% of the US population, to provide a sub-state national analysis of power outages affecting Americans. Over 70% of study counties experienced at least one 8+ hour outage during the study period. Medically relevant 8+ hour outages were prevalent in the South, Northeast, and Appalachia and some areas on the West Coast, while 1+ hour outages patterned similarly with a higher

**Table 2 | County-day co-occurrence of severe weather or climate events and 8+ hour outages**

| Severe weather or climate event | County-days, N (%) | 8+ h outage county-days, N (%) | Co-occurrence Ratio[a] |
|---|---|---|---|
| Total | 1,799,319 (100.0) | 22,793 (100.0) | — |
| None | 1,265,213 (70.2) | 8637 (37.9) | – |
| Isolated event[b] | 492,489 (27.4) | 11,310 (49.7) | 3.4 |
| Heavy precipitation | 267,823 (14.9) | 8507 (37.3) | 4.7 |
| Snowfall | 25,523 (1.4) | 1172 (5.1) | 6.7 |
| Anomalous heat | 131,727 (7.3) | 1090 (4.8) | 1.2 |
| Anomalous cold | 57,924 (3.2) | 344 (1.5) | 0.9 |
| Wildfire | 5381 (0.3) | 84 (0.4) | 2.3 |
| Lightning | 3578 (0.2) | 63 (0.3) | 2.6 |
| Tropical cyclone | 533 (0.03) | 50 (0.2) | 13.7 |
| Multiple event[c] | 41,617 (2.3) | 2846 (12.5) | 10.0 |
| Heavy precipitation-anomalous heat | 17,415 (1.0) | 917 (4.0) | 7.7 |
| Heavy precipitation-cyclone | 2650 (0.2) | 679 (3.0) | 37.5 |
| Heavy precipitation-lightning | 8142 (0.5) | 644 (2.8) | 11.6 |
| Snowfall-anomalous cold | 7151 (0.1) | 250 (1.1) | 5.1 |
| Heavy precipitation-anomalous cold | 1779 (0.1) | 134 (0.6) | 11.0 |
| Heavy precipitation-cyclone-anomalous heat | 244 (0.0) | 86 (0.4) | 51.6 |
| Heavy precipitation-anomalous heat-lightning | 1197 (0.1) | 78 (0.3) | 9.5 |
| Other | 3039 (0.2) | 58 (0.3) | 2.8 |

Analysis included the 1653 continental counties with 3 years of reliable data. We defined co-occurrence if the weather/climate event occurred in the same county on the same day the 8+ hour outage began. We separated snowfall out from heavy precipitation.

[a]The co-occurrence ratio was the proportion of county-days with severe weather or climate event type *i* that co-occurred with an 8+ hour outage divided by the proportion of county-days without any weather or climate event that co-occurred with an 8+ hour outage. A co-occurrence ratio > 1 means that 8+ hour outages were more likely to occur on county-days with severe weather or climate event *i* compared to days with no event and a ratio <1 means that 8+ hour outages were less likely to occur on county-days with severe weather or climate event *i* compared to days with no event.

[b]An isolated event was a county-day where only a single severe weather or climate event occurred. We defined a county exposed to lightning if a lightning flash happened, tropical cyclone if the county was within 100 km of a cyclone path, wildfire if the county intersects a ≥ km² wildfire, heat if temperatures exceed 24 °C and that is above the 85th percentile, cold if temperatures are below 0 °C and below the 15th percentile, snowfall if snow accumulation > 1 inch, and heavy precipitation if precipitation > 85th percentile.

[c]Multiple events indicate county-days where multiple severe weather or climate events occurred. We include the top 7 most common multiple event types (among power outage days) plus all other multiple events grouped and provide the full breakdown in Supplemental Table 3. Power outage data was purchased from PowerOutage.us.

concentration in the South. Most outages co-occurred with severe weather or climate events, particularly heavy precipitation, anomalous heat, and tropical cyclones. We observed clusters of counties facing both frequent 8+ hour outages and high social and medical vulnerability measured by SVI and Medicare DME use prevalence. Louisiana and Arkansas had many counties with high 8+ hour outage-high SVI and high 8+ hour outage-high DME use prevalence.

Few studies have evaluated power outage exposure nationally. Prior studies have used US Department of Energy data and characterized "electric emergency incidents and disturbances" at the state-level. This included outages affecting 50,000+ customers or an unplanned loss of 300 MW. Most such outage events affected coastal states[4,17]. The EIA reports yearly summaries of annual outage interruptions at the state level and found that 2020 had the longest average durations of outage events[2]. Our county-level study evaluating national power outage exposure at a sub-state geographic scale contributes to the growing literature on outages as an environmental health exposure.

We used commercial data from PowerOutage.us to generate relative metrics that accounted for differences in county customer counts and an absolute metric that based on total annual customer hours without power. Both metrics have utility but provide different information. Relative metrics describe disparities and are commonly used to evaluate health inequities, while absolute metrics measure the total burden of exposure in a population[20]. A strength of our study is that we provide both types of metrics for use in a range of contexts, from health studies to policy to emergency preparedness and management.

We observed that most county-days with an 8+ hour outage coincided with one or more severe weather or climate events. Prior national state-level studies reported that storms and severe weather led to over 50% of outages[1,17]. When evaluating blackouts, outages affecting 50,000+ customers or a loss of 300 MW, Hines et al. found that wind/rain-driven blackouts increased between 1984–2006 across all US regions[21]. As severe weather and climate events increase with climate change, this trend is likely to continue[22]. Our study identified specific event types most likely to co-occur with 8+ hour outages regionally and seasonally, with heavy precipitation (year-round) and snowfall (winter) predominating in the Northeast, Midwest, and South and snowfall (winter) and wildfire (fall) leading in the West. Utilities, customers, and policymakers could use this information for planning and resource allocation. We also found that nearly 40% of county-days with 8+ hour outages occurred without one of the included severe weather or climate event that our study considered. These cases were likely due to technical problems such as equipment failure or transmission delays[17]. Although weather and climate events seem to drive recent large-scale outages, issues with the aging electrical grid and increases in demand remain, both of which may be contributors to smaller-scale outages at the county level.

A robust literature has established that environmental exposures such as air pollution and drinking water violations disproportionately affect certain groups such as low-income, communities of color, and under-resourced groups[23,24]. However, the environmental justice literature has not equally engaged with power outages, power restoration, or their possible inequitable distribution. Prior energy justice studies have noted that natural disasters can accentuate disparities in power outages and restoration. For example, power restoration time reflects which communities are prioritized and by extension which communities are neglected. In Puerto Rico after Hurricane Maria, Sotolongo et al. observed that rural and Black communities experienced the longest restoration times[25], and Tormos-Aponte et al. found that social vulnerability and political marginalization were linked to longer wait times for the arrival of restoration crews[26]. During the

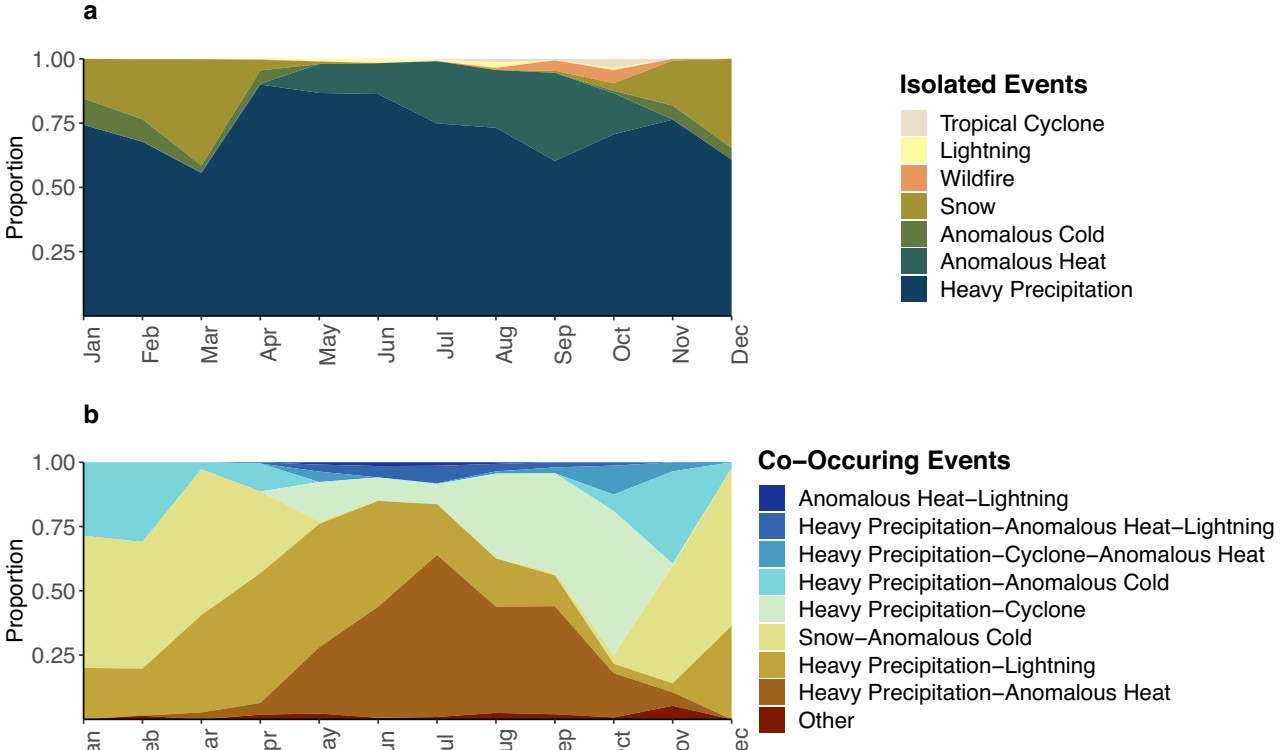

**Fig. 5 | Monthly distribution of severe weather or climate events on days they co-occurred with 8 + hour outages among counties with 3 years of data.** **a** Isolated severe weather and climate events (*n* = 11,310 county-days). **b** Multiple severe weather and climate event combinations (*n* = 2,846 county-days). An isolated event was a county-day where only a single severe weather or climate event occurred and multiple events were county-days where more than one event occurred. We defined a county exposed to lightning if a lightning flash happened, tropical cyclone if the county was within 100 km of a cyclone path, wildfire if the county intersects a ≥ km² wildfire, anomalous heat if temperatures exceed 24 °C and that is above the 85th percentile, anomalous cold if temperatures are below 0 °C and below the 15th percentile, snowfall if snow accumulation > 1 inch, and heavy precipitation if precipitation > 85th percentile. Power outage data was purchased from PowerOutage.us.

**Table 3 | Distribution of 1 + hour and 8 + hour outages by quartile of SVI and prevalence of Medicare DME users per 1000**

| Metric | Quartile | Quartile values | Total count, *N* | | Median (IQR) count, *N* | | Max count, *N* | |
|---|---|---|---|---|---|---|---|---|
| | | | 1+ hour | 8+ hour | 1+ hour | 8+ hour | 1+ hour | 8+ hour |
| SVI | Q1 | [0–0.28] | 22,468 | 1185 | 40 (73) | 1 (4) | 195 | 25 |
| | Q2 | (0.28–0.53] | 28,228 | 1405 | 57 (92) | 2 (4) | 217 | 22 |
| | Q3 | (0.53–0.77] | 34,911 | 1678 | 68 (99) | 2 (5) | 330 | 26 |
| | Q4 | (0.77–1] | 40,600 | 1888 | 80 (83) | 3 (5)[a] | 414 | 35 |
| DME use, per 1000 Medicare enrollees | Q1 | [0–45] | 32,211 | 1616 | 62 (95) | 2 (5) | 234 | 26 |
| | Q2 | (45–58] | 33,565 | 1704 | 63 (98) | 2 (5) | 394 | 35 |
| | Q3 | (58–74] | 35,276 | 1765 | 68 (98) | 2 (5) | 414 | 25 |
| | Q4 | (74–478] | 25,155 | 1071 | 42 (82) | 1 (3)[a] | 217 | 22 |

Analysis includes 2038 counties with 2+ years of reliable data.

[a]The Wilcoxon rank sum test *p*-value < 0.01 when comparing the median of 8+ hour power outages in (1) counties belonging to the highest SVI (social vulnerability index) quartile versus all other counties and (2) counties belonging to the highest DME (durable medical equipment) use quartile versus all other counties. Power outage data was purchased from PowerOutage.us.

Texas winter storm in 2021, Flores et al. observed that counties with a higher proportion of Hispanic/Latino residents faced more severe outages and that Black individuals reported more day-long outages via questionnaires[15]. In Florida after Hurricane Irma, higher percentages of Hispanic/Latino populations were associated with longer outages[14]. Our nationwide study found significantly higher median annual counts of 1+ and 8+ hour outages in high versus low SVI counties.

Studies have previously incorporated SVI into research about COVID-19, heat exposure, and hurricanes[27–29]. Flanagan et al. demonstrated SVI's utility in informing policy and intervention for those most affected by disaster events like Hurricane Katrina[28]. Our study identified counties that experienced high 8+ hour outage exposure and high

SVI, largely concentrated in Louisiana, Mississippi, Arkansas, and Michigan. These counties versus all others had significantly higher proportions of residents living in poverty, of "racial/ethnic minority status," and living in mobile homes, making them potentially more vulnerable to disasters and likely in need of more resources to deal with such events. For example, the co-occurrence of natural disasters and outages has the potential to exacerbate adverse health outcomes among vulnerable communities. This can occur in the case of co-occurring anomalous temperatures and outages where disadvantaged groups may have worse baseline health, lower access to generators, higher occupational exposures, and more urban heat island effect exposure[4,30]. Further, New Yorkers living in public housing after

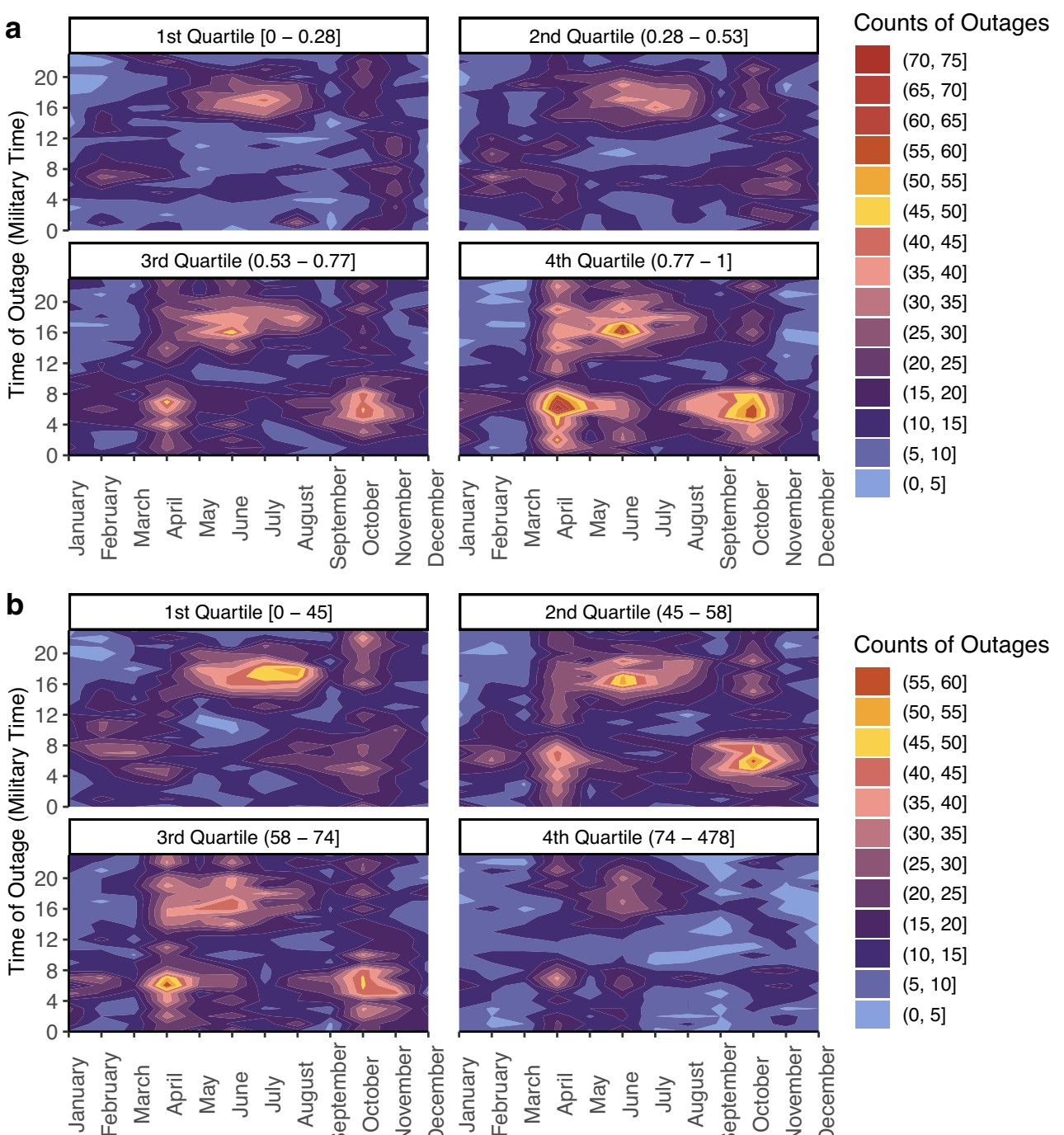

**Fig. 6 | Contour plot of the count of 8+ hour outages according to start time of outage per month by county SVI and DME use quartile.** Analysis includes 2038 counties with 2+ years of reliable data. **a** SVI quartile category. **b** Medicare DME use per 1000 Medicare enrollees quartile category. DME, durable medical equipment; SVI, social vulnerability index. Power outage data was purchased from PowerOutage.us.

Hurricane Sandy reported inability to purchase basic necessities because of widespread power outages in stores[7]. Outages shaped some participants' decision to remain in place despite evacuation warnings because people worried about personal safety and property theft[7]. Communities with high vulnerability face a particular set of concerns during outage events, which have implications for their health, so it is important to allocate resources to these communities to support them during outage events.

We found lower median annual 8+ hour outage exposure (1 versus 2) in the highest quartile of Medicare DME use compared to other quartiles. Despite this trend, it is crucial to consider that DME users are

particularly susceptible to the health consequences of outages. Prior research showed that during outages, emergency rooms saw a higher proportion of DME users seeking care and hospitals needed to make external referrals for treatment[31,32]. Medicare DME users may be especially vulnerable because they are either older adults or individuals with disabilities. Longer outages especially endanger DME users due to possible limited battery life of equipment. For example, typical battery life ranges from 3–4 h for oxygen concentrators on the lowest settings[10]. Emergency planning guidance tends to place the onus on DME users to adequately prepare[33]. A study in Michigan found that only a quarter of older adults using essential electricity-dependent

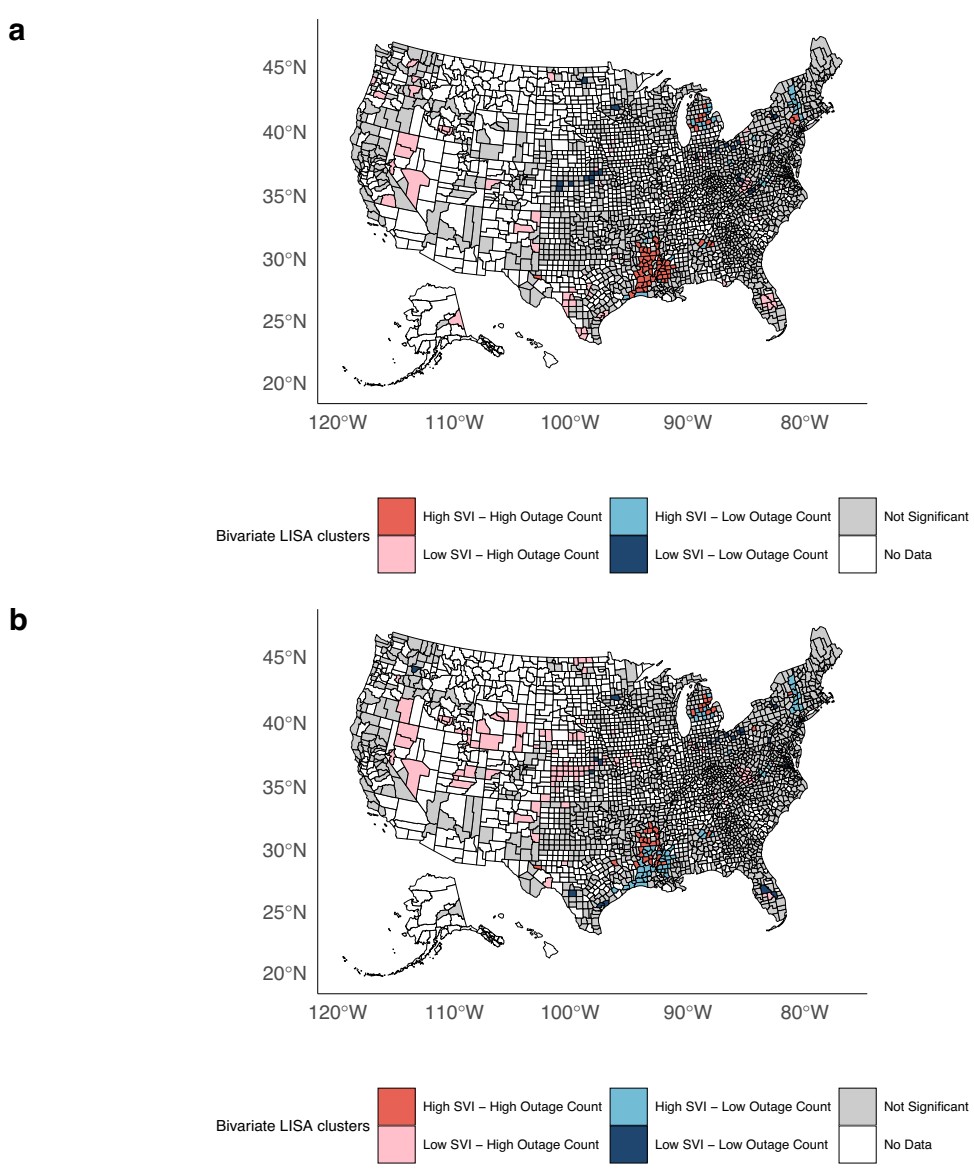

**Fig. 7 | Bivariate Local Spatial Clustering Analysis (LISA) of 8+ hour outages and SVI and DME use quartiles.** The analysis includes 2038 counties. **a** SVI and 8+ hour outage LISA. Counties in red indicate high SVI and high 8+ hour outage counts. **b** Medicare DME users and 8+ hour outage LISA. Counties in red indicate high prevalence of Medicare DME users and high 8+ hour outage counts. DME, durable medical equipment; SVI, social vulnerability index. Power outage data was purchased from PowerOutage.us and county basemaps were obtained from the usmap R package version 0.6.1.

medical equipment had an alternative power source[34]. In New York City, those with electricity-dependent household members had lower perceived preparedness (32%) than those without a DME user at home (47%)[35]. Data from California suggests that DME use prevalence and equipment rental days have increased over time, particularly among lower SES individuals (e.g., users of Medicaid)[36]. While we did not identify disproportionate exposure to 8+ hour outages among Medicare DME users, given the growing Medicare DME population, their high vulnerability to power outages, and our findings of geographic clustering of high 8+ hour outage exposure and high concentrations of DME users, emergency preparedness officials should prioritize this demographic in planning[37].

Our study identified clusters of counties that experienced high 8+ hour outage exposure and were either socially or medically vulnerable. Knowledge about vulnerability can inform equitable disaster preparedness and response, and several organizations have begun efforts to collect it. Though not designed explicitly for power outages, the

California Office of Environmental Health and Hazard Assessment created the California Communities Environmental Health Screening Tool (CalEnviroScreen) to identify specific communities most affected by social stressors and various sources and forms of pollution[38]. The tool's purpose is to inform and guide regulations with environmental justice in mind. We used other vulnerability metrics, the CDC's SVI and the Department of Health and Human Services' emPOWER dataset, which are part of larger efforts to identify vulnerable communities with the goal of disaster preparation[37,39]. Prior studies leveraged medical records from hospital databases to map the location of DME users in Massachusetts and Centers for Medicaid and Medicare Service's list of individuals using oxygen concentrators or ventilators in New Orleans[40], proposing that there are publicly available data and sources for informing disaster preparedness as well[22,36]. While social vulnerability is increasingly integrated into disaster management for equitable emergency response, it lacks a major role in grid investment strategies. Initial work has integrated social vulnerability into micro-

grid strategies, considering critical infrastructure locations under different outage scenarios[41]. Our results point to US counties where medically and socially vulnerability overlaps with high outage burden, information that could guide investments to reduce societal burdens from outages among the most vulnerable.

Our study had limitations, several related to the PowerOutage.us data. Not all US utilities appeared in the dataset, with small rural utilities most often absent. Due to a combination of the proportion of customer coverage and temporal missingness, we lacked reliable data on 563 counties, many of which were in the Midwest and Mountain West. However, our report still represents the most comprehensive, county-level summary of power outages to date, covering 2447 (78.9%) US counties. Our data spanned only 3 years, so we could not evaluate long-term trends. We assessed power outages at the county-level, which did not account for sub-county heterogeneity in exposure. For example, a county-level outage could occur due to either several minor sub-county outages or a single large outage at one sub-county location. Further, our county-aggregated data did not ensure the same customers were without power during 1+ and 8+ hour outages. For example, in one outage definition, we required 0.1% of total county customers to be without power for 8+ hours, but the composition of the 0.1% without power could change during the outage. Spatio-temporal granularity is necessary for accurate outage exposure measurement, so future studies, particularly those interested in linking outages to individual health outcomes, should consider exposure at a sub-county geographic resolution such as the household or building level, perhaps using improved power utility data, internet-connected devices, or satellite imagery[42–46]. Finally, we identified county-days where severe weather and climate events co-occurred with outages, but due to data limitations, we could not causally link severe weather and climate events to outages. Future studies that have access to cause-specific outage data would add to this growing literature.

There were also limitations with selected vulnerability metrics. As with the power outage data, county-level measures of vulnerability may have masked sub-county level trends. While we found a correlation between county-level outages and high SVI, it is possible this relationship would differ with finer-scale data. SVI is a summed rank of many vulnerability factors, which comprehensively describes county composition but may also include factors less relevant to power outage vulnerability. The metric is constructed to identify counties vulnerable to disasters but not designed specifically for power outages. Future studies may be interested in evaluating individual sociodemographic characteristics or other metrics at finer spatial resolutions. Regarding DME use prevalence, the emPOWER dataset undercounts total DME users as it only covers Medicare recipients. However, DME use increases with age and disability, and older adults and individuals with disabilities are eligible for Medicare, so we likely capture the majority of DME use[36].

Despite health consequences of outages, few studies have characterized their duration, geographic distribution, linkage to weather/climate events, or exposure disparities. Policymakers and public health and emergency preparedness officials need this data to equitably allocate resources to communities most burdened by and vulnerable to outage events. Our county-level power outage exposure data could also support future large-scale epidemiology studies[4], as we continue to learn more about the health effects of these primarily climate-driven events. The absolute and relative power outage metrics generated herein can inform future policy about electricity and healthcare infrastructure planning in the face of climate change.

## Methods
### Ethics statement
The Columbia University Institutional Review Board approved this research (Protocol #AAAT5765).

**Power outages and utility customers data.** From PowerOutage.us, we purchased 10 min resolution power outage information, which included the number of customers without power and the time of reporting from 2017–2020 for counties in all US states. Due to the low spatial coverage for 2017, we a priori excluded this year from analyses, so the study spanned 2018–2020. Customers refers to residential consumers such as families and non-residential consumers such as businesses. PowerOutage.us gathered outage data at subcounty levels (e.g., cities census-designated places) at regular 10 min intervals using utility providers' application programming interfaces (API).

We summarized the coverage of our power outage data by utility type using information from the US Energy Information Administration (EIA). The EIA tracks information about power usages for each US state and should theoretically record all operating utility providers. They have annual data on utility providers by state and service providers by county. Our power outage data captured cooperatives (59.4%), which typically serve rural communities, and most investor-owned utilities (84.4%), which tend to serve large populations.

To generate our outage dataset, we aggregated PowerOutage.us data to the county and hourly level. Of the 3,142 US counties, PowerOutage.us reported some data from 3010 (95.8%). We completed data quality and reliability checks and removed unreliable counties from certain analyses (Fig. 1). Broadly, we considered county APIs to be reliably reporting on outages if the APIs report ≥50% of the time, and we consider an API to reliably capture customers within a county if reported customers covered ≥50% of total county customers (Supplementary Methods 1.1 and 1.2). After applying these criteria, 2447 counties remained, covering 73.7% of the US population. Most analyses focused on the 2,038 counties with 2+ years of reliable data.

**Power outage metrics: power outage event and outage experience definitions.** We set out to define a binary power outage variable for each county-hour (1 = power out, 0 = power on). To do so, we had to define a threshold of customers without power over which we considered the county to be experiencing an outage. This relative power outage event definition accounted for county total customers and enabled us to compare outage counts across counties with varying population sizes.

We defined a power outage event as occurring whenever the percent of customers without power met or exceeded 0.1% of the county customers. Counts of customers without power came from PowerOutage.us data, and we estimated the total number of county customers based on households from the 2015–2019 American Community Survey and business establishments from the 2021 Census Bureau (Supplementary Methods 1.1). If county A had 100,000 customers, it met our power outage definition when 100 customers or more lost power; county B, with 1 million customers, would require at least 1000 customers to lose power to meet the definition. We used a 0.1% threshold corresponding to the 90th percentile of customers out per hour at the county-level during the study period. Prior studies have used the 90th percentile to determine outage events[35,42].

We computed the duration of outage events as the total time a county's percent of customers without power continuously reached or exceeded the 0.1% threshold (Supplementary Figure 1). We considered outages of 8+ hour and 1+ hour duration. Outages of 1+ hour duration would disrupt commerce and other activities, and 8+ hour outages would likely impact health by surpassing critical thresholds, including the maximum battery life for certain DME[10]. To summarize 8+ hour and 1+ hour outages, we took the average of the total number of events annually by county over the study period. This metric is similar to the System Average Interruption Frequency Index (SAIFI)[47].

We also calculated an absolute outage metric: annual average county-level customers without power. This metric identifies counties where the greatest absolute count of customers experienced loss of power. Because county-level customer density differs dramatically, we

also computed a second absolute metric: annual average county-level number of minutes without power per customer. This metric is similar to the System Average Interruption Duration Index (SAIDI)[47]. It can be interpreted as minutes without power experienced by the average customer in a county.

Characterizing power outage exposure – especially when investigating disparities – necessitates both relative and absolute metrics. Our relative metric for outage events accounts for county customer density so that we may compare across counties. Our absolute metrics for outage experiences identify counties with the highest count of affected customers and the geographic distribution of total time without power.

**Severe weather and climate event data and definitions.** We identified the following severe weather and climate events at the daily-county-level: anomalous heat/cold, heavy precipitation, snowfall, lightning, tropical cyclones, and wildfires from a variety of data sources (Supplementary Methods 2). Data sources included the Parameter-elevation Regressions on Independent Slopes Model (temperature, precipitation), the National Gridded Snowfall Analysis (snowfall), the International Space Station Lightning Imaging Sensor (lightning), the International Best Track Archive for Climate Stewardship project (tropical cyclones), and the National Interagency Fire Center (wildfires). We defined a county as exposed to an anomalous heat event if the temperature exceeded 24 °C and was above the 85th percentile of weekly temperatures from 1981–2010, an anomalous cold event if temperatures dipped below 0 °C and was below the 15th percentile of weekly temperatures from 1981–2010, heavy precipitation if daily precipitation exceeded the 85th county percentile, snowfall if 2.54 cm (1″) or above of snow accumulation occurred, a lightning event if a lightning flash occurred within a county, a tropical cyclone if the county boundary was within 100 km of a tropical cyclone track center, and a wildfire if the county intersected with a $\geq 1\,km^2$ wildfire. We identified days with a single, isolated event and days with multiple events separately as more severe and co-occurring weather and climate events likely cause more damage to the electrical grid than single events. We did not include wind since wind and precipitation were previously observed to be highly correlated[48].

**Vulnerability data and definitions.** Those relying on electricity-dependent DME require constant access to electricity to maintain and manage their health. This vulnerability means power outages can rapidly worsen health conditions and increase mortality risk. To characterize this group, we generated county-level prevalence of DME use among Medicare enrollees using the December 2020 emPOWER dataset from the US Department of Health & Human Services. We calculated quartiles of DME use prevalence per 1000 Medicare beneficiaries in each county for analysis.

Another group vulnerable to the consequences of outages are disadvantaged communities requiring extra support before, during, and after disasters. To identify such counties, we used the US Centers for Disease Control and Agency for Toxic Substances and Disease Registry's Social Vulnerability Index (SVI). SVI has the stated purpose to identify specific areas that may need additional disaster-related support. Such information about a county's overall social vulnerability can shape decisions about future preparedness strategies or resource allocation, particularly in the event of longer outages that may affect health. SVI is based on 16 census variables from the 2016–2020 American Community Survey (below 150% poverty, unemployed, housing cost burden, no high school diploma, no health insurance, aged 65 and older, aged 17 and younger, civilian with a disability, single-parent households, English language proficiency, racial and ethnic minority status, multi-unit structures, mobile homes, crowding, no vehicle, and group quarters), which are used to create 4 vulnerability themes (socioeconomic status, household composition &

disability, minority status & language, housing type & transportation). SVI ranges from 0 to 1 where county indices closer to 0 indicate lower vulnerability and indices closer to 1 indicate higher vulnerability. We generated quartiles of SVI for analysis.

**Statistical analysis.** Initial analyses were descriptive leveraging all reliable counties in our dataset ($N = 2447$), reporting the frequency of outages by county and total and average customer-hours without power. To better understand the relationship between severe weather and climate events and outages, we identified county-days where severe weather and climate events co-occurred with 8+ hour outage events. We summarized this information by month. Because this was a daily analysis and weather data was available for the continental US, we used only continental counties with 3 full years of data ($n = 1653$). Additionally, we calculated a co-occurrence ratio of severe weather and climate events with 8+ hour outages. The co-occurrence ratio was computed as the proportion of county-days with severe weather or climate event type $i$ that co-occurred with an 8+ hour outage divided by the proportion of county-days without any weather or climate event that co-occurred with an 8+ hour outage. A co-occurrence ratio >1 means that 8+ hour outages were more likely to occur on county-days with severe weather or climate event $i$ compared to days with no event and a ratio <1 means that 8+ hour outages were less likely to occur on county-days with severe weather or climate event $i$ compared to days with no event.

We then conducted Wilcoxon Rank Sum tests to evaluate the relationship between county-level annual averages of 8+ hour outage counts for counties with 2+ years of reliable data ($n = 2038$) and DME use prevalence and SVI (vulnerability metrics). To identify spatial clusters of counties with both high outage exposure and high vulnerability, we ran separate bivariate local indicators of association (LISA) analyses, two-sided tests[49]. The bivariate analyses capture the relationship between the value of our vulnerability metric at one county location and the spatial lag of 8+ hour outages in surrounding counties. We used the rgeoda package version 0.0.9 to run 99,999 permutations, set the cluster significance at $\alpha = 0.05$, and applied the false discovery rate method to correct for multiple comparison testing[50]. Because the bivariate LISA conducts multiple hypothesis testing for each county, the probability of spurious statistical significance increases. Applying the false discovery rate is recommended and limits spurious positive findings[51,52]. We investigated whether the underlying components of SVI differed between high-outage-high SVI county clusters and all others and used two-sided t-tests to evaluate whether they differed statistically. All analyses were conducted in R version 4.1.0 (2021-05-18). Code used to run the bivariate LISA can be found on GitHub at https://github.com/viviando/National-Power-Outages[53]. We used the usmap R package version 0.6.1 to generate our nationwide maps for this study[54]. The package uses the U.S Census Bureau cartographic boundary files and the Albers equal-area conic projection.

**Reporting summary**
Further information on research design is available in the Nature Portfolio Reporting Summary linked to this article.

## Data availability
The power outage data that support the findings of this study are available for purchase from PowerOutage.us at https://PowerOutage.us/products. Processed data containing annual average counts of outage events and customers without power are available at https://github.com/viviando/National-Power-Outages. The Centers for Disease Control and Prevention Social Vulnerability Index data is available publicly at https://www.atsdr.cdc.gov/placeandhealth/svi/data_documentation_download.html and the Health and Human Services Medicare durable medical equipment data is available for download at https://empowerprogram.hhs.gov/empowermap. A full description of

data used to generate severe weather and climate events is available in Supplementary Information Methods 2.

## Code availability

The code for analysis can be found at the GitHub repository[55]: https://doi.org/10.5281/zenodo.7668274.

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

## Acknowledgements

This work was funded by the National Institute on Aging (NIA) grant RF1AG071024 [JAC], National Institute of Environmental Health Sciences (NIEHS) grant P30ES009089 [JAC], and NIEHS grant 2T32ES007322 [VD]. The authors would like to thank Benjamin Steiger and Milo Gordon at the Columbia Mailman School of Public Health for their support in processing the wildfire data.

## Author contributions

V.D. made contributions to the conception/design of the work, the analysis and interpretation of data, and draft of the work. She approves the submitted version and agrees both to be personally accountable and to ensure that questions related to the accuracy or integrity of any part of the work are appropriately resolved. H.B. made contributions to the interpretation of data and draft of the work. She approves the submitted version and agrees both to be personally accountable and to ensure that questions related to the accuracy or integrity of any part of the work are appropriately resolved. N.M.F. made contributions to the interpretation of data and draft of the work. She approves the submitted version and agrees both to be personally accountable and to ensure that questions related to the accuracy or integrity of any part of the work are appropriately resolved. A.J.N. made contributions to the interpretation of data and draft of the work. He approves the submitted version and agrees both to be personally accountable and to ensure that questions related to the accuracy or integrity of any part of the work are appropriately resolved. M.V.K. made contributions to the interpretation of data and draft of the work. He approves the submitted version and agrees both to be personally accountable and to ensure that questions related to the accuracy or integrity of any part of the work are appropriately resolved. J.S. made contributions to the interpretation of data and draft of the work. He approves the submitted version and agrees both to be personally accountable and to ensure that questions related to the accuracy or integrity of any part of the work are appropriately resolved. J.A.C. made contributions to the conception/design of the work, the analysis and interpretation of data, and draft of the work. She approves the submitted version and agrees both to be personally accountable and to ensure that questions related to the accuracy or integrity of any part of the work are appropriately resolved.

## Competing interests

The authors declare no competing interests.
