## [Peer Review File · Nature Communications]

Spatiotemporal distribution of power outages with climate events and social vulnerability in the USAREVIEWER COMMENTS

Reviewer #2 (Remarks to the Author):

The authors should be commended for moving the scholarship forward on power outage exposure. Their effort demonstrates a clearer conceptual framework regarding the interaction of extant social and health vulnerabilities and outage exposure than previously published, and provides numerous opportunities for other points of departure.

However, the piece suffers from some fairly serious internal validity and construct validity threats that should be discussed more thoroughly in the paper in order to contextualize the findings appropriately as well as provide the authors with more clarity about how to proceed. Historical and instrumentation are perhaps the most significant internal validity threats. On the former, the 2018-2020 timeframe of the collected data coincides with several major hazard events (the Ranch, Lake, and Soledad fires in wealthy, white sections of Los Angeles County, CA and Tropical Storm Imelda across Harris County, TX to name some clear examples associated with the authors' chosen highlights). These events occurring in such a short analysis time distort the associations that the author's seek to explore. A more appropriate timeframe would be at least 10 years, including major events like these to better demonstrate the pattern. On the latter, the use of SVI (which uses ACS 2014-2018 data) is also problematic as it does not track accurately with the nature of the defined vulnerability measure over the time or conditions observed. In fact, changes in any of the underlying SVI constructs could be confused with a treatment effect.

Another challenging threat is the threat to the definition and measurement of the vulnerability construct. The county-level aggregate construct which is SVI is neither geographically aligned with the authors' conceptualization of outage exposures' health effects (which should be measured at much more granular geography), nor does it reveal much about the individual components' relevant contributions to the aggregate construct of vulnerability or their internal consistency (eg, their intercorrelation). The geographic problem can't be addressed immediately without major additional data collection. As the authors could foreground more, outage is best measured at a household level, but the outage data purchased are at the county level—meaning the resulting exposure-vulnerability association is weak. Census tract-level data would be better if that were possible for purchase to avoid PII but still provide more accurate demographic and related vulnerability factor (eg housing quality and inadequacy) variation. [Side note: most utilities do report outages online at the property level! They just do this in real-time and won't provide it historically without negotiation. So, researchers could conceivable machine-scrape these data as they occur, record them over time, and then analysis at a level of aggregation that could still give nuance to vulnerabilities.]

Beyond the geographic challenge, though is the explication issue which could be pushed further without too much effort: the underlying components of SVI are readily available at the county-level. The authors should conduct a few tests for these against the outage exposure observations--that is, describe/ correlate/ regress on these too rather than the SVI. They could also possibly run a test of their convergent validity to the authors' other primary medical vulnerability variable (the DME counts).

On the subject of construct validity, though, I should add that the underlying outage construct's operationalization was well done given its data limitations--and that is something that I'm sure the authors will continue to play with as further data become available. More geographically granular data would allow for a more discrete definition of outage (ie below the current 0.5% county pop measure to a combined absolute threshold (eg min 100 residential units in a census tract) and proportional threshold (over 5% of a tract)).

Ultimately, the work elicits much thinking on this subject, but the conclusions and interpretations are not supported because of the data limitations. When that happens, a work like this could still be publishable. It just needs a lot of disclaimers and at least a bit more substantive heft (such as the

more nuanced vulnerability component analysis).

A few thoughts about taking the work further beyond these fixes and more robust disclaimers and discussions of limitations:

- if more granular geographic data are not available, extending the timeframe of their analysis would reveal more reliable patterns about vulnerability (but this would also require using the appropriate ACS data for the periodicity and not SVI).
- the authors note several documented health effects from outages, but don't include obvious ones such as hypothermia and heat stroke that occur seasonally or temporarily. These hazards relate to lack of adequate heating, cooling and other housing inadequacies that may be associated with some of the primary vulnerability components.
- for many of the above health effects, data of realized effects are available at the county-level. Certainly, mortality, asthma-related hospital visits, etc. Would it be worthwhile to compare actual outages with realized health effects rather than outages with possible health and social vulnerability?
- there are non-health effects from outages that become later health hazards, such as financial and employment losses, food insecurity, social cohesion breakdown, etc. The authors might look at these as well.
- because the PowerOutage.US data is available at a granular temporal level (every 10 minutes), it might help to assess whether there are seasonal (ie, winter or summer) or temporal (night and day) differences in outage effects on vulnerability or realized health hazards. These could be controlled, or might prove a whole new set of research questions.
- the authors might consider adding a case study of a place that includes both geographically granular outage, vulnerability, and health outcome data as well as qualitative analysis of households' responses to the outages. The Los Angeles cases might be ideal because CA has good granular health data, and LA DWP would likely share its outage info for research purposes.

A few minor quibbles:

- The threshold operationalization is strong but one of the outcome measures (customer-hours without power) needs to be weighted by population. CA and TX will always have the biggest absolute numbers of everything. The authors note this construct is "aligned with population size" but that can be addressed easily.
- API isn't defined. Do you mean the American Petroleum Institute or Associated Press?

Reviewer #3 (Remarks to the Author):

Thank you for the opportunity to review this piece. This is an important contribution to a growing literature on energy insecurity and intersecting dimensions of health, climate, and disaster vulnerability. This article offers an expansive and novel view on power outages and develops two key measures to compare outages: "the relative power outage event defined by the proportion of county customers without power and the absolute customer hour totals" (268-269). The process of data collection and methodology of the article are sound and synthesize several bodies of social science, health, and energy literatures to create a measure for the intensity and incidence of power outages. The analysis supports the conclusions that the authors assert, however, there are a few areas in which I would recommend additional discussion to strengthen the contributions of the article.

First, on a methodological level, I was curious whether the authors sought to account for effects of longer power outages such as those linked to specific disasters such as major hurricanes or floods. In other words, do your conclusions show outages in relatively "normal" conditions and reveal a background rate of power outages, or do they correspond to major weather disasters in particular? Was there any effort to account for outlier events such as Hurricane Michael that might skew the outages in Georgia in 2018 for example? The findings stand on their own without completing another dimension of analysis, but this would be useful to note in either the methods or discussion section. Second, I felt that the authors buried some of the most important findings of this piece in the discussion and conclusions section. Perhaps expand the introduction to specify why a comparative

measure of power outage is needed (line 82-87) for health studies. The stakes of this are not referenced until the discussion where you note the vulnerability of DME-users and the individual health and social costs of outages. Is there a way to foreshadow these findings in the introduction to give the reader more context to understand the methods section?

Third, there is a robust literature on energy justice that investigates the inequitable distribution of power in many cases from a starting point of environmental justice. You cite Diana Hernández, and other scholars whose work might be relevant here include Tony Reames, Shalanda Baker, and Sanya Carley. I would suggest formatting your discussion of existing environmental justice literature on lines 279-280 to focus on power outages, which have been underexamined relative to other processes of energy distribution and access, or alternatively you could engage some of the more recent literature in this area. I would suggest:

- Sotolongo, M., L. Kuhl, and S. H. Baker. 2021. Using environmental justice to inform disaster recovery: Vulnerability and electricity restoration in Puerto Rico. *Environmental Science & Policy* 122:59–71.

- Tormos-Aponte, F., G. García-López, and M. A. Painter. 2021. Energy inequality and clientelism in the wake of disasters: From colorblind to affirmative power restoration. *Energy Policy* 158:112550.

Finally, please reference the attached file where I have highlighted several syntax and grammatical questions. Notably, I would recommend changing “the Appalachia” to Appalachia.

Thank you again for this important contribution. I look forward to seeing your work in print.

REVIEWER COMMENTS

Reviewer #2 (Remarks to the Author):

The authors should be commended for moving the scholarship forward on power outage exposure. Their effort demonstrates a clearer conceptual framework regarding the interaction of extant social and health vulnerabilities and outage exposure than previously published, and provides numerous opportunities for other points of departure.

However, the piece suffers from some fairly serious internal validity and construct validity threats that should be discussed more thoroughly in the paper in order to contextualize the findings appropriately as well as provide the authors with more clarity about how to proceed.

Thank you for your comments. We provide responses below.

- 1. Historical and instrumentation are perhaps the most significant internal validity threats. On the former, the 2018-2020 timeframe of the collected data coincides with several major hazard events (the Ranch, Lake, and Soledad fires in wealthy, white sections of Los Angeles County, CA and Tropical Storm Imelda across Harris County, TX to name some clear examples associated with the authors' chosen highlights). These events occurring in such a short analysis time distort the associations that the author's seek to explore. A more appropriate timeframe would be at least 10 years, including major events like these to better demonstrate the pattern.**

We agree that a longer timeframe would be informative to assess trends and long-term patterns across the US. Unfortunately, our data is limited to 2018-2020 but provides a snapshot of power outages in the US at that time. In response to concerns regarding natural hazards, we now include additional analyses on 1,653 counties with 3 years of reliable data where we evaluate the co-occurrence of 8+ hour outages and severe storms and climate events across the nation and for each census region. We find high overlap between severe weather and climate events with 8+ hour outages and add text and results to the manuscript in several sections detailing methods and results.

Abstract:

- “62.1% of the 8+ hour outages co-occurred with an extreme weather or climate event.”

Summary paragraph, introduction:

- “62.1% of the 8+ hour outages co-occur with an extreme weather or climate event and 8+ hour outages are 3.4x more common on days with a single event and 10x more common on days with multiple events.” (Lines 232-234).

Included “**Severe Weather and Climate Event Data and Definitions**” in Methods that details the data used to generate our severe weather and climate event variables

For our statistical analysis, we calculate a co-occurrence ratio:

- “The co-occurrence ratio was computed as the proportion of county-days with severe weather or climate event type *i* that co-occurred with an 8+ hour outage divided by the proportion of county-days without any weather or climate event that co-occurred with an 8+ hour outage. A co-occurrence ratio > 1 means that 8+ hour outages were more likely to occur on county-days with severe weather or climate event *i* compared to days with no event and a ratio < 1 means that 8+ hour outages were less likely to occur on county-days with severe weather or climate event *i* compared to days with no event.” (Lines 446-464)

Results on the co-occurrence of severe weather and climate events and 8+ hour outages are presented under the heading “Severe weather and climate events and 8+ hour outages” in the results section as well as in several main and supplementary tables/figures as follows:

- **Figure 5.** Monthly distribution of severe weather or climate events on days they co-occurred with 8+ hour outages among counties with 3 years of data. a Isolated severe weather and climate events (N = 11,310 county-days). b Multiple severe weather and climate event combinations (N = 2,846 county-days).

- **Table 2.** County-day co-occurrence of severe weather or climate events and 8+ hour outages.
 - **Supplementary Figure 4.** Census region monthly distribution of severe weather or climate events on days they co-occurred with 8+ hour outages among counties with 3 years of data.
 - **Supplementary Table 4.** County-day co-occurrence of severe weather or climate events and 8+ hour outages, including all combinations of multiple events.
2. **On the latter, the use of SVI (which uses ACS 2014-2018 data) is also problematic as it does not track accurately with the nature of the defined vulnerability measure over the time or conditions observed. In fact, changes in any of the underlying SVI constructs could be confused with a treatment effect.**

Another challenging threat is the threat to the definition and measurement of the

vulnerability construct. The county-level aggregate construct which is SVI is neither geographically aligned with the authors' conceptualization of outage exposures' health effects (which should be measured at much more granular geography), nor does it reveal much about the individual components' relevant contributions to the aggregate construct of vulnerability or their internal consistency (eg, their intercorrelation). The geographic problem can't be addressed immediately without major additional data collection.

We appreciate the reviewer's concern regarding SVI and note the thoughtful points on the weaknesses of the SVI construct. However, we chose the index because its stated purpose is "to help emergency response planners and public health officials identify and map communities that will most likely need support before, during, and after a hazardous event."² Because we consider outages – particularly 8+ hour outages with implications for human health – as hazardous events, we used SVI as a tool to provide information for policy and resource allocation decisions. We clarify our intentions with the following:

- "Another group vulnerable to the consequences of outages are disadvantaged communities requiring extra support before, during, and after disasters. To identify such counties, we used the U.S. Centers for Disease Control and Agency for Toxic Substances and Disease Registry's Social Vulnerability Index (SVI). SVI has the stated purpose to identify specific areas that may need additional disaster-related support. Such information about a county's overall social vulnerability can shape decisions about future preparedness strategies or resource allocation, particularly in the event of longer outages that may affect health." (Lines 409-428)

In the prior submission we used the available 2014-2018 SVI data, but this next version has been released in the interim and we now update results using the 2016-2020 SVI, indicating this vulnerability metric is

- "based on 16 census variables from the 2016-2020 American Community Survey". (Lines 428-429)

We also add a sentence to the limitations regarding the temporal mismatch of SVI and outages:

- "While we found a correlation between county-level outages and high SVI, it is possible this relationship would differ with finer-scale data. SVI is a summed rank of many vulnerability factors, which comprehensively describes county composition but may also include factors less relevant to power outage vulnerability. The metric is constructed to identify counties vulnerable to disasters but not designed specifically for power outages. Future studies may be interested in evaluating individual sociodemographic characteristics or other metrics at finer spatial resolutions." (Lines 1042-1050)

3. **As the authors could foreground more, outage is best measured at a household level, but the outage data purchased are at the county level—meaning the resulting exposure-vulnerability association is weak. Census tract-level data would be better if that were possible for purchase to avoid PII but still provide more accurate demographic and related vulnerability factor (eg housing quality and inadequacy) variation. [Side note: most utilities do report outages online at the property level! They just do this in real-time and won't provide it historically without**

negotiation. So, researchers could conceivable machine-scrape these data as they occur, record them over time, and then analysis at a level of aggregation that could still give nuance to vulnerabilities.]

We agree that household-level data is ideal, particularly for the study of power outages and health. We are also aware of several ongoing efforts to scrape data in the way described by the reviewer, but these data cover only a few utilities and do not offer multi-year timescales (yet). The data currently available from PowerOutage.US were best summarized at the county-level. We now further acknowledge the ideal spatial scale of outage exposure by including the following sentence in the limitations section of our study:

- “Spatiotemporal granularity is necessary for accurate outage exposure measurement, so future studies, particularly those interested in linking outages to individual health outcomes, should consider exposure at a sub-county geographic resolution such as the household or building level, perhaps using improved power utility data, internet-connected devices, or satellite imagery.²⁻⁵” (Lines 1031-1035)

4. Beyond the geographic challenge, though is the explication issue which could be pushed further without too much effort: the underlying components of SVI are readily available at the county-level. The authors should conduct a few tests for these against the outage exposure observations--that is, describe/ correlate/ regress on these too rather than the SVI. They could also possibly run a test of their convergent validity to the authors' other primary medical vulnerability variable (the DME counts).

We agree understanding the relation between specific components of SVI (e.g., percent living in poverty) could be a second important analysis. We chose to use the full index to identify counties at risk before, during, and after a hazardous event. Single elements of SVI likely do not collectively identify vulnerable counties as well, so we keep our main analyses related to the full SVI. However, we agree that understanding which characteristics are most prevalent in high SVI counties that also experience a high prevalence of outages is of interest. So, we now examine individual vulnerability components by characterizing the distribution of the 16 SVI components (e.g., Below 150% Poverty, Unemployed, Housing Cost Burden) for the counties experiencing high 8+ hour outage and high SVI. Results of this analysis appear in the text of the results and supplementary information:

- “Among high outage-high SVI counties compared to all others, the components of SVI contributing to a high SVI score were higher percentages of “racial and ethnic minority” individuals (40.4% vs. 24.0%), individuals living below 150% poverty (34.0% vs. 24.3%), and those living in mobile homes (20.1% vs. 12.4%). T-tests also showed that these differences were statistically significant (p-value < 0.05) (Supplementary Table 5).” (Lines 723-727)
 - **Supplementary Table 5.** Attributes of counties with high SVI and high 8+ hour outage exposure.
- 5. On the subject of construct validity, though, I should add that the underlying outage construct's operationalization was well done given its data limitations--and that is something that I'm sure the authors will continue to play with as further data become available. More geographically granular data would allow for a more discrete definition**

of outage (ie below the current 0.5% county pop measure to a combined absolute threshold (eg min 100 residential units in a census tract) and proportional threshold (over 5% of a tract)).

Thank you for this point. Indeed, we continue to work on improving the operationalization of outage exposure and have begun to obtain more spatially granular data in certain states. At smaller-scale geographic units, setting a floor for residential units or applying a person-time without power over a defined interval may prove most fruitful. The present manuscript provides customer-hours without power as a secondary metric to our binary Y/N 1-hour or 8-hour outage counts. We now add a sentence to the discussion to this point:

- “We used commercial data from PowerOutage.US to generate relative metrics that accounted for differences in county customer counts and an absolute metric that based on total annual customer hours without power. Both metrics have utility but provide different information. Relative metrics describe disparities and are commonly used to evaluate health inequities, while absolute metrics measure the total burden of exposure in a population.⁶ A strength of our study is that we provide both types of metrics for use in a range of contexts, from health studies to policy to emergency preparedness and management.” (Lines 781-787)

Additionally, we added a sentence that emphasizes the importance for future studies to consider the necessity of spatial granularity for outage exposure measurement, particularly for health research:

- “Spatiotemporal granularity is necessary for accurate outage exposure measurement, so future studies, particularly those interested in linking outages to individual health outcomes, should consider exposure at a sub-county geographic resolution such as the household or building level, perhaps utilizing improved power utility data, internet-connected devices, or satellite imagery.²⁻⁵” (Lines 1031-1035).

- 6. Ultimately, the work elicits much thinking on this subject, but the conclusions and interpretations are not supported because of the data limitations. When that happens, a work like this could still be publishable. It just needs a lot of disclaimers and at least a bit more substantive heft (such as the more nuanced vulnerability component analysis). A few thoughts about taking the work further beyond these fixes and more robust disclaimers and discussions of limitations:
- if more granular geographic data are not available, extending the timeframe of their analysis would reveal more reliable patterns about vulnerability (but this would also require using the appropriate ACS data for the periodicity and not SVI).**

We appreciate the reviewer acknowledging the limitations of the data and opportunities for us to improve our manuscript. Unfortunately, data is not available from PowerOutage.US prior to 2017. We do have 2017 data, but the spatial coverage is so low that we had to exclude it from analysis *a priori*. We have gone through the manuscript carefully and added qualifiers about limitations. We also now much more clearly define assumptions made to generate our dataset and put new requirements into place on temporal data coverage. This includes an extensive supplementary methods section detailing decisions made to improve transparency and reproducibility. The following changes were made:

- “Due to the low spatial coverage for 2017, we *a priori* excluded this year from analyses, so the study spanned 2018-2020. Customers refers to residential consumers such as families and non-residential consumers such as businesses.” (Lines 246-248)
- “To generate our outage dataset, we aggregated PowerOutage.US data to the county and hourly level. Of the 3,142 US counties, PowerOutage.US reported some data from 3,010 (95.8%). We completed data quality and reliability checks and removed unreliable counties from certain analyses (Figure 1). Broadly, we considered county APIs to be reliably reporting on outages if the APIs report $\geq 50\%$ of the time, and we consider an API to reliably capture customers within a county if reported customers covered $\geq 50\%$ of total county customers (Supplementary Methods 1.1 and 1.2). After applying these criteria, 2,447 counties remained, covering 73.7% of the US population. Most analyses focused on the 2,038 counties with 2+ years of reliable data.” (Lines 264-271)
- “Our study had limitations, several related to the PowerOutage.US data. Not all US utilities appeared in the dataset, with small rural utilities most often absent. Due to a combination of the proportion of customer coverage and temporal missingness, we lacked reliable data on 563 counties, many of which were in the Midwest and Mountain West. However, our report still represents the most comprehensive, county-level summary of power outages to date, covering 2,447 (78.9%) US counties. Our data spanned only 3 years, so we could not evaluate long-term trends. We assessed power outages at the county-level, which did not account for sub-county heterogeneity in exposure.” (Lines 1019-1026)
- See **Supplementary Methods 1**, which thoroughly describes assumptions
- See **Figures 3 and 4**, which now include information on years of reliable data available by U.S. counties included in our analyses.
- The variables used to generate the SVI are not available at the 1-year resolution so we could not evaluate yearly changes in SVI. Thus, we assessed differences between the 2014-2018 SVI and 2016-2020 SVI. The **extra figure 1** below demonstrates the county score difference between SVI 2020 and SVI 2018. The absolute SVI differences for most counties (72.2%) were less than 0.1, suggesting that overall, the SVI measure remains similar. Further, the rank order of counties in terms of SVI was also quite stable between 2018 and 2020 (see **extra figure 2** below).

Extra figure 1: County SVI score difference between 2020 SVI and 2018 SVI.

Extra figure 2: County SVI rank order between 2018 and 2020.

7. the authors note several documented health effects from outages, but don't include obvious ones such as hypothermia and heat stroke that occur seasonally or temporarily.

These hazards relate to lack of adequate heating, cooling and other housing inadequacies that may be associated with some of the primary vulnerability components.

We appreciate the opportunity to rectify this oversight. We have included the following sentences:

- “Documented health effects include carbon monoxide poisoning from improper generator use, anxiety, stress, and exacerbation of existing cardiovascular and respiratory conditions.⁷ Because outages can prevent the use of temperature-controlling devices, risk of hypothermia and heatstroke can increase when outages occur during extreme cold spells and heatwaves.⁸ Moreover, outages can lead to acute food insecurity when refrigerators lack power⁹, fear related to personal safety¹⁰, and economic losses in commercial and industrial sectors¹¹.” (Lines 133-138)

The new sentence builds on an already existing sentence to illustrate the connection among outages, temperature-related illnesses, and social vulnerability:

- “Others vulnerable to power outages include under-resourced communities and historically marginalized groups. Pathways include disrupted hourly employment, older and less-insulated housing stock resulting in dangerous indoor temperatures, lack of access to cooling facilities, and a higher burden of underlying chronic diseases sensitive to extreme temperatures.^{12,13}” (Lines 142-146)

8. for many of the above health effects, data of realized effects are available at the county-level. Certainly, mortality, asthma-related hospital visits, etc. Would it be worthwhile to compare actual outages with realized health effects rather than outages with possible health and social vulnerability?

Thank you for this point. The publicly-available health data proposed are typically annual in scale, while the hypothesized health effects of outages are likely predominately acute. We currently have studies underway that evaluate health outcomes and other vulnerability metrics at an improved spatiotemporal resolution, but these data do not have the geographic coverage of the present analysis. In the present paper, we aimed to first characterize the nationwide distribution of outages, linkages with extreme weather and climate events, and their overlap with vulnerabilities (both social and medical), as no such analyses have been published to date. The hope is that the present manuscript can provide the basis for future studies linking outage exposure to health outcomes.

9. there are non-health effects from outages that become later health hazards, such as financial and employment losses, food insecurity, social cohesion breakdown, etc. The authors might look at these as well.

We have included the following sentence in the manuscript to acknowledge potentially broader pathways through which outages affect health:

- “Moreover, outages can lead to acute food insecurity when refrigerators lack power,⁹ fear related to personal safety,¹⁰ and economic losses in commercial and industrial

sectors¹¹” (Lines 137-138).

- 10. because the PowerOutage.US data is available at a granular temporal level (every 10 minutes), it might help to assess whether there are seasonal (ie, winter or summer) or temporal (night and day) differences in outage effects on vulnerability or realized health hazards. These could be controlled, or might prove a whole new set of research questions.**

We thank the reviewer for identifying potential future avenues of research. The present analysis has three main domains: nationwide characterization of outages using (1) relative and absolute metrics, (2) linkages with severe weather and climate events, and (3) overlap with medical and social vulnerability. We do provide seasonal information on domains (1), (2), and (3) finding seasonality and within day temporality in outage start times as well as different co-occurring weather and climate events by month and geographic region. Domain (3) geographic region and seasonal data is presented in Figure 6 and Supplementary Figure 6. Future health studies will certainly wish to evaluate seasonal and temporal differences in health effects and if these relations are modified by social vulnerability. We have ongoing work in this space.

- 11. the authors might consider adding a case study of a place that includes both geographically granular outage, vulnerability, and health outcome data as well as qualitative analysis of households' responses to the outages. The Los Angeles cases might be ideal because CA has good granular health data, and LA DWP would likely share its outage info for research purposes.**

Thank you for this comment and proposal. As the reviewer has accurately pointed out, health studies benefit from high spatiotemporal outage data, and our group has several ongoing projects linking subcounty and sub-daily outage metrics to sub-daily health outcomes.

A few minor quibbles:

- 12. The threshold operationalization is strong but one of the outcome measures (customer-hours without power) needs to be weighted by population. CA and TX will always have the biggest absolute numbers of everything. The authors note this construct is "aligned with population size" but that can be addressed easily.**

We initially included customer-hours without power to identify regions with the largest number of people without power because we felt this absolute number could inform where policy dollars and planning services could be targeted. We agree that this metric is partially driven by population density and now also include the reviewer's suggestion of customers weighted by population and included the following:

- “Because county-level customer density differs dramatically, we also computed a second absolute metric: annual average county-level number of minutes without power per customer.” (Lines 369-371)

Results appear as Panel 4b in the main manuscript:

- **Figure 4. County yearly averages of customers without power. Counties shaded in white lacked any reliable data.**

13. API isn't defined. Do you mean the American Petroleum Institute or Associated Press?

We have clarified that API refers to application programming interface:

- “PowerOutage.US gathered outage data at subcounty levels (e.g., cities, census designated places) at regular 10-minute intervals using utility providers’ application programming interfaces (API).” (Line 248-255)

Reviewer #3 (Remarks to the Author):

1. **Thank you for the opportunity to review this piece. This is an important contribution to a growing literature on energy insecurity and intersecting dimensions of health, climate, and disaster vulnerability. This article offers an expansive and novel view on power outages and develops two key measures to compare outages: “the relative power outage event defined by the proportion of county customers without power and the absolute customer hour totals” (268-269). The process of data collection and methodology of the article are sound and synthesize several bodies of social science, health, and energy literatures to create a measure for the intensity and incidence of power outages. The analysis supports the conclusions that the authors assert, however, there are a few areas in which I would recommend additional discussion to strengthen the contributions of the article.**

Thank you for your feedback and encouragement.

2. **First, on a methodological level, I was curious whether the authors sought to account for effects of longer power outages such as those linked to specific disasters such as major hurricanes or floods. In other words, do your conclusions show outages in relatively “normal” conditions and reveal a background rate of power outages, or do they correspond to major weather disasters in particular? Was there any effort to account for outlier events such as Hurricane Michael that might skew the outages in Georgia in 2018 for example? The findings stand on their own without completing another dimension of analysis, but this would be useful to note in either the methods or discussion section.**

We found this point important, and in this revision, we have added an analysis investigating the co-occurrence of severe weather/climate events and 8+ hour outages. We limited this analysis to the 1,653 continental counties with 3 years of data where we evaluate the co-occurrence of 8+ hour outages and severe storms and climate events across the nation and for each census region.

We find high overlap between severe weather and climate events with 8+ hour outages and add text and results to the manuscript in several sections detailing methods and results.

Abstract:

- “62.1% of the 8+ hour outages co-occurred with an extreme weather or climate event.”

Summary paragraph, introduction:

- “62.1% of the 8+ hour outages co-occur with an extreme weather or climate event and 8+ hour outages are 3.4x more common on days with a single event and 10x more common on days with multiple events.” (Lines 232-234).

Included “**Severe Weather and Climate Event Data and Definitions**” in Methods that details the data used to generate our severe weather and climate event variables

For our statistical analysis, we calculate a co-occurrence ratio:

- “The co-occurrence ratio was computed as the proportion of county-days with severe weather or climate event type *i* that co-occurred with an 8+ hour outage divided by the proportion of county-days without any weather or climate event that co-occurred with an 8+ hour outage. A co-occurrence ratio > 1 means that 8+ hour outages were more likely to occur on county-days with severe weather or climate event *i* compared to days with no event and a ratio < 1 means that 8+ hour outages were less likely to occur on county-days with severe weather or climate event *i* compared to days with no event.” (Lines 446-464)

Results on the co-occurrence of severe weather and climate events and 8+ hour outages are presented under the heading “Severe weather and climate events and 8+ hour outages” in the results section as well as in several main and supplementary tables/figures as follows:

- **Figure 5.** Monthly distribution of severe weather or climate events on days they co-occurred with 8+ hour outages among counties with 3 years of data. a Isolated severe weather and climate events (N = 11,310 county-days). b Multiple severe weather and climate event combinations (N = 2,846 county-days).

- **Table 2.** County-day co-occurrence of severe weather or climate events and 8+ hour outages.
- **Supplementary Figure 4.** Census region monthly distribution of severe weather or climate events on days they co-occurred with 8+ hour outages among counties with 3 years of data.
- **Supplementary Table 4.** County-day co-occurrence of severe weather or climate events and 8+ hour outages, including all combinations of multiple events.

3. **Second, I felt that the authors buried some of the most important findings of this piece in the discussion and conclusions section. Perhaps expand the introduction to specify why a comparative measure of power outage is needed (line 82-87) for health studies. The stakes of this are not referenced until the discussion where you note the vulnerability of DME-users and the individual health and social costs of outages. Is there a way to foreshadow these findings in the introduction to give the reader more context to understand the methods section?**

We have included the following sentence to foreshadow the importance of both measures of power outage:

- “The relative metric accounts for population size, while the absolute metric identifies counties with the largest count of customers without power. Both metrics provide important information about which counties to prioritize for intervention and resource allocation, especially in the context of social and medical vulnerabilities.” (Lines 222-225)
4. **Third, there is a robust literature on energy justice that investigates the inequitable distribution of power in many cases from a starting point of environmental justice. You cite Diana Hernández, and other scholars whose work might be relevant here include Tony Reames, Shalanda Baker, and Sanya Carley. I would suggest formatting your discussion of existing environmental justice literature on lines 279-280 to focus on power outages, which have been underexamined relative to other processes of energy distribution and access, or alternatively you could engage some of the more recent literature in this area. I would suggest:**
 - Sotolongo, M., L. Kuhl, and S. H. Baker. 2021. Using environmental justice to inform disaster recovery: Vulnerability and electricity restoration in Puerto Rico. *Environmental Science & Policy* 122:59–71.
 - Tormos-Aponte, F., G. García-López, and M. A. Painter. 2021. Energy inequality and clientelism in the wake of disasters: From colorblind to affirmative power restoration. *Energy Policy* 158:112550.

Thank you for the suggested literature. We have made the following edits to engage with more recent literature in the area and to further root our discussion in existing environmental justice literature on outages:

- “However, the environmental justice literature has not equally engaged with power outages, power restoration, or their possible inequitable distribution. Prior energy justice studies have noted that natural disasters can accentuate disparities in power outages and restoration. For example, power restoration time reflects which communities are prioritized and by extension which communities are neglected. In Puerto Rico after Hurricane Maria, Sotolongo et al. observed that rural and Black communities experienced the longest restoration times¹⁴, and Tormos-Aponte et al. found that social vulnerability and political marginalization were linked to longer wait times for the arrival of restoration crews.¹⁵ During the Texas winter storm in 2021, Flores et al. observed that counties with a higher proportion of Hispanic residents faced more severe outages and that Black individuals reported more day-long outages via questionnaires.¹⁶ In Florida after Hurricane Irma, higher percentages of Hispanic and Latino populations

were associated with longer outages.¹⁷ Our nationwide study found significantly higher median annual counts of 1+ and 8+ hour outages in high versus low SVI counties.”
(Lines 944-957)

- 5. Finally, please reference the attached file where I have highlighted several syntax and grammatical questions. Notably, I would recommend changing “the Appalachia” to Appalachia.**

Thank you for these edits. We have implemented the changes, including replacing “the Appalachia” with “Appalachia” in the manuscript.

- 6. Thank you again for this important contribution. I look forward to seeing your work in print.**

Thank you for your encouragement.

REFERENCES IN RESPONSE DOCUMENT

1. Agency for Toxic Substances & Disease Registry. CDC/ATSDR’s Social Vulnerability Index (SVI). Published April 28, 2021. Accessed January 11, 2022.
<https://www.atsdr.cdc.gov/placeandhealth/svi/index.html>
2. Zhang W, Sheridan SC, Birkhead GS, et al. Power Outage. *Chest*. 2020;158(6):2346-2357. doi:10.1016/j.chest.2020.05.555
3. Meier A, Ueno T, Pritoni M. Using data from connected thermostats to track large power outages in the United States. *Applied Energy*. 2019;256:113940. doi:10.1016/j.apenergy.2019.113940
4. Min B, O’Keeffe Z, Zhang F. *Whose Power Gets Cut? Using High-Frequency Satellite Images to Measure Power Supply Irregularity*. World Bank, Washington, DC; 2017. doi:10.1596/1813-9450-8131
5. Levin N, Kyba CCM, Zhang Q, et al. Remote sensing of night lights: A review and an outlook for the future. *Remote Sensing of Environment*. 2020;237:111443. doi:10.1016/j.rse.2019.111443
6. King NB, Harper S, Young ME. Use of relative and absolute effect measures in reporting health inequalities: structured review. *BMJ*. 2012;345(sep03 1):e5774-e5774. doi:10.1136/bmj.e5774
7. Casey JA, Fukurai M, Hernández D, Balsari S, Kiang MV. Power Outages and Community Health: a Narrative Review. *Curr Envir Health Rpt*. 2020;7(4):371-383. doi:10.1007/s40572-020-00295-0

8. Cheshire WP. Thermoregulatory disorders and illness related to heat and cold stress. *Autonomic Neuroscience*. 2016;196:91-104. doi:10.1016/j.autneu.2016.01.001
9. Hecht AA, Biehl E, Buzogany S, Neff RA. Using a trauma-informed policy approach to create a resilient urban food system. *Public Health Nutr*. 2018;21(10):1961-1970. doi:10.1017/S1368980018000198
10. Hernández D, Chang D, Hutchinson C, et al. Public Housing on the Periphery: Vulnerable Residents and Depleted Resilience Reserves post-Hurricane Sandy. *J Urban Health*. 2018;95(5):703-715. doi:10.1007/s11524-018-0280-4
11. LaCommare KH, Eto JH. *Understanding the Cost of Power Interruptions to U.S. Electricity Consumers*. Lawrence Berkeley National Lab. (LBNL), Berkeley, CA (United States); 2004. doi:10.2172/834270
12. Lewis J, Hernández D, Geronimus AT. Energy efficiency as energy justice: addressing racial inequities through investments in people and places. *Energy Efficiency*. 2020;13(3):419-432. doi:10.1007/s12053-019-09820-z
13. Hernández D. Understanding ‘energy insecurity’ and why it matters to health. *Social Science & Medicine*. 2016;167:1-10. doi:10.1016/j.socscimed.2016.08.029
14. Sotolongo M, Kuhl L, Baker SH. Using environmental justice to inform disaster recovery: Vulnerability and electricity restoration in Puerto Rico. *Environmental Science & Policy*. 2021;122:59-71. doi:10.1016/j.envsci.2021.04.004
15. Tormos-Aponte F, García-López G, Painter MA. Energy inequality and clientelism in the wake of disasters: From colorblind to affirmative power restoration. *Energy Policy*. 2021;158(C). Accessed September 30, 2022. <https://ideas.repec.org/a/eee/enepol/v158y2021ics0301421521004201.html>
16. Flores NM, McBrien H, Do V, Kiang MV, Schlegelmilch J, Casey JA. The 2021 Texas Power Crisis: distribution, duration, and disparities. *J Expo Sci Environ Epidemiol*. Published online August 13, 2022;1-11. doi:10.1038/s41370-022-00462-5
17. Mitsova D, Esnard AM, Sapat A, Lai BS. Socioeconomic vulnerability and electric power restoration timelines in Florida: the case of Hurricane Irma. *Nat Hazards*. 2018;94(2):689-709. doi:10.1007/s11069-018-3413-x

REVIEWERS' COMMENTS

Reviewer #2 (Remarks to the Author):

As a previous reviewer of this submission, I am pleased to see the consideration that the authors have towards providing the appropriate disclaimers around the study's generalizability given the limitations of the data which they analyze. While the original data still severely limit analysis of the hypotheses the authors seek to explore, these disclaimers provide sufficient warning and context to the reader.

The additions of references and the other corrections are appreciated as well.

Reviewer #3 (Remarks to the Author):

Thank you for the thorough comments and response to the reviewers. The authors have addressed all of my concerns from the original version of this paper. This is an important intervention to understand the geographies of power outages and health implications of these events. The authors acknowledge the limitations of their data and analysis, and point to important areas for further inquiry and application of this work.

REVIEWERS' COMMENTS

Reviewer #2 (Remarks to the Author):

As a previous reviewer of this submission, I am pleased to see the consideration that the authors have towards providing the appropriate disclaimers around the study's generalizability given the limitations of the data which they analyze. While the original data still severely limit analysis of the hypotheses the authors seek to explore, these disclaimers provide sufficient warning and context to the reader.

The additions of references and the other corrections are appreciated as well.

We appreciate your comments about our updates in the first round of revisions.

Reviewer #3 (Remarks to the Author):

Thank you for the thorough comments and response to the reviewers. The authors have addressed all of my concerns from the original version of this paper. This is an important intervention to understand the geographies of power outages and health implications of these events. The authors acknowledge the limitations of their data and analysis, and point to important areas for further inquiry and application of this work.

Thank you for your encouragement for this project.